# Unraveling the molecular mechanism of MIL-53(Al) crystallization

Daniil Salionov [1], Olesya O. Semivrazhskaya[2], Nicola P. M. Casati[3], Marco Ranocchiari [4], Saša Bjelić[1], René Verel [5], Jeroen A. van Bokhoven [4,5✉] & Vitaly L. Sushkevich [4✉]

The vast structural and chemical diversity of metal—organic frameworks (MOFs) provides the exciting possibility of material's design with tailored properties for gas separation, storage and catalysis. However, after more than twenty years after first reports introducing MOFs, the discovery and control of their synthesis remains extremely challenging due to the lack of understanding of mechanisms of their nucleation and growth. Progress in deciphering crystallization pathways depends on the possibility to follow conversion of initial reagents to products at the molecular level, which is a particular challenge under solvothermal conditions. The present work introduces a detailed molecular-level mechanism of the formation of MIL-53(Al), unraveled by combining in situ time-resolved high-resolution mass-spectrometry, magic angle spinning nuclear magnetic resonance spectroscopy and X-ray diffraction. In contrast to the general belief, the crystallization of MIL-53 occurs via a solid-solid transformation mechanism, associated with the spontaneous release of monomeric aluminum. The role of DMF hydrolysis products, formate and dimethylamine, is established. Our study emphasizes the complexity of MOF crystallization chemistry, which requires case-by-case investigation using a combination of advanced in situ methods for following the induction period, the nucleation and growth across the time domain.

[1] Bioenergy and Catalysis Laboratory, Paul Scherrer Institute, 5232 Villigen PSI, Switzerland. [2] Laboratory for Organic Chemistry, ETH Zürich, Vladimir-Prelog-Weg 3, 8093 Zürich, Switzerland. [3] Laboratory for Synchrotron Radiation - Condensed Matter, Paul Scherrer Institute, 5232 Villigen PSI, Switzerland. [4] Laboratory for Catalysis and Sustainable Chemistry, Paul Scherrer Institute, 5232 Villigen PSI, Switzerland. [5] Institute for Chemistry and Bioengineering, ETH Zurich, Vladimir-Prelog-Weg 1, 8093 Zurich, Switzerland. ✉email: jeroen.vanbokhoven@chem.ethz.ch; vitaly.sushkevich@psi.ch

Metal-organic frameworks (MOFs) are solid materials comprising multi-functionalized organic molecules bound to metal ions or clusters through coordinating moieties, such as carboxylates, imidazoles, amines, sulfates, and phosphates, to form microporous and mesoporous crystalline compounds possessing one-, two-, and three-dimensional structures[1,2]. Their topology can be changed by varying the connectivity of the organic linker and/or the inorganic unit. The perceptible advantage of MOFs over materials, such as zeolites and other porous oxides, is their larger surface area, larger pore diameter, and chemical versatility that result from their composition and crystalline structure. However, it took almost twenty years to extrapolate this peculiarities to areas such as catalysis[3–6] and gas separation and storage[7–12]. Equally, MOFs can function as support for metal or oxide nanoparticles, finding application in photovoltaics and electrocatalysis[13,14].

There are numerous examples showing the engineering of novel MOFs at the molecular level[15]. However, even after several decades of research, the field of understanding of the synthesis of MOFs remains a gray area due to the absence of insight to the mechanisms of their nucleation and crystallization[16–20]. The lack of this knowledge hinders the rational design of MOF for target applications[21–23]. Unraveling the synthesis mechanism of solid compounds is of enormous interest to both academia and industry due to the possibility to engineer and then synthesize materials with specific properties[23,24]. Needless to say, studying the crystallization processes of MOFs is a complex endeavor due to the heterogeneity of the synthesis medium, comprising a solvent, inorganic precursors, organic linkers, modulators, templates, etc. and high sensitivity to various physical and chemical parameters, such as temperature, pressure, pH, the composition of the reaction mixture, and aging time[25–28].

Nucleation and growth from soluble, pre-formed, secondary building units (SBUs) or related molecular structures[29] are among the most accepted mechanisms established, for instance, for the family of aluminum trimesates of MIL-96, MIL-100, and MIL-110 topology[30]. According to this model, the crystallization of MOF starts with the formation of SBU, comprising metal clusters or ions, bound to the organic linker. Next, they aggregate and nucleate, forming the seeds for further crystallization. The transport of SBUs or/and molecular precursors from the liquid phase to the seeds finalizes the crystallization of the material, which can subsequently undergo Ostwald ripening, resulting in larger crystals. Simultaneously, more elementary mechanisms, such as growing from oligomers or pre-formed clusters, have also been suggested in the case of UiO-66[31,32]. In contrast, the evolution of amorphous nanoparticles, observed during the crystallization of MIL-89(Fe) and developing into ordered structures, constitutes an alternative to the liquid transport crystallization model[33,34]. Crucially, none of these models explicitly explains the frequent observation of intermediate phases, such as MOF-235 or EHU-30, formed at the early stages of the solvothermal synthesis[35,36]. This issue is highlighted in the recent reviews, urging the researchers to develop and use in situ methods for monitoring the synthesis[21,22].

There are indeed quite a number of research papers devoted to the in situ analysis of the formation of MOFs under solvothermal conditions, mostly using X-ray scattering techniques, which are intrinsically sensitive to the observation of crystalline phases, and, in some cases, to the amorphous nuclei[16,35,37]. Equally, a very limited number of reports is devoted to the application of NMR[30,38], Raman[39], X-ray absorption spectroscopy[33], or mass spectrometry[40], which makes the overall area of the understanding synthesis extremely underexplored. Moreover, no single technique can interrogate the many species that exist during MOF synthesis on timescales from seconds to days, but each

chemical reaction and process during framework construction affects the product that forms and its properties. NMR spectroscopy can provide the information on the kinetics of nucleation, being, however, intrinsically blind to the induction period due to the low resolution and signal broadening. In contrast, high-resolution mass spectrometry is ideal for the direct establishing the structure of soluble building units and pre-nucleation species. X-ray diffraction (XRD) is of natural choice to follow the formation of the crystalline phase of target MOF or its by-products. A combination of various techniques becomes essential to explore the synthesis in the time domain from induction period through nucleation to final precipitation, hence enabling the refinement of the current crystallization models for different synthetic protocols.

Aluminum-based MOFs are especially important due to their high thermal stability (>773 K) and relatively low synthesis cost. MIL-53, one of the most studied MOFs, is of particular interest due to its breathing behavior upon adsorption of guest molecules[40]. The formation of this thermodynamically stable phase is frequently observed during the synthesis of other MOFs, such as MIL-101, MIL-68, and MOF-235[35,39,41]. In this work, we focused our attention on studying MIL-53(Al) crystallization mechanism using in situ electrospray ionization coupled to high-resolution mass spectrometry (ESI-HRMS), magic angle spinning NMR and X-ray diffraction. To do so, we have developed the methodology, which can become a compelling approach when studying the mechanism of the synthesis of (in)organic functional materials. The combination of physical-chemical tools, data processing approach, and discovered mechanistic concepts pave a road towards understanding the synthesis mechanism and engineering properties of metal-organic frameworks, with broad implications in fields ranging from adsorbents and catalysts to photovoltaics and semiconductors.

## Results

**High-resolution mass spectrometry.** The chemical processes and reactions leading to the formation of the final MOF start immediately after the mixing of the initial reagents aluminum nitrate and terephthalic acid ($H_2BDC$) dissolved in N,N-dimethylformamide (DMF). Figure 1A, B shows the exemplary ESI-HRMS spectra of the synthetic mixture after 100 min of reaction at 343 K acquired in negative and positive modes, respectively. Using isotope tracing, the majority of the most intense peaks were assigned (for details, see Supporting Information, Tables 1, 2, Figs. S1–S4). The species containing up to four aluminum atoms were identified in the range of 200–1000 $m/z$, coordinated to the ligands $NO_3^-$, $BDC^{2-}$, $HDBC^-$, DMF, $OH^-$, $OCH_3^-$, and $HCOO^-$. The first four originate from the initial synthetic mixture, while the latter are coming either from the hydrolysis of aluminum nitrate nonahydrate and N,N-dimethylformamide or coordination of methanol, which is used as a solvent for ESI-HRMS analysis. The structure of the aluminum-containing species does not vary significantly in negative and positive modes—the main difference is associated with the coordination of one or two DMF molecules to each positively charged specie, which it is not the case for anions. At a longer reaction time, the formation of dimethylammonium nitrate salts $((CH_3)_2NH_2)_x(NO_3)_{x-1}^+$ was observed in positive mode. Comparison of the time-resolved responses of the normalized intensities of each identified MS peak enabled the grouping of the peaks in the classes with very similar temporal behavior. Three main groups comprise the following compounds:

(i) *monomeric Al*, which has only one aluminum atom in the structure, charge-balanced with $NO_3^-$, $OH^-$, $OCH_3^-$, and DMF. None of the aluminum coordinated to terephthalic acid species follows a similar behavior.

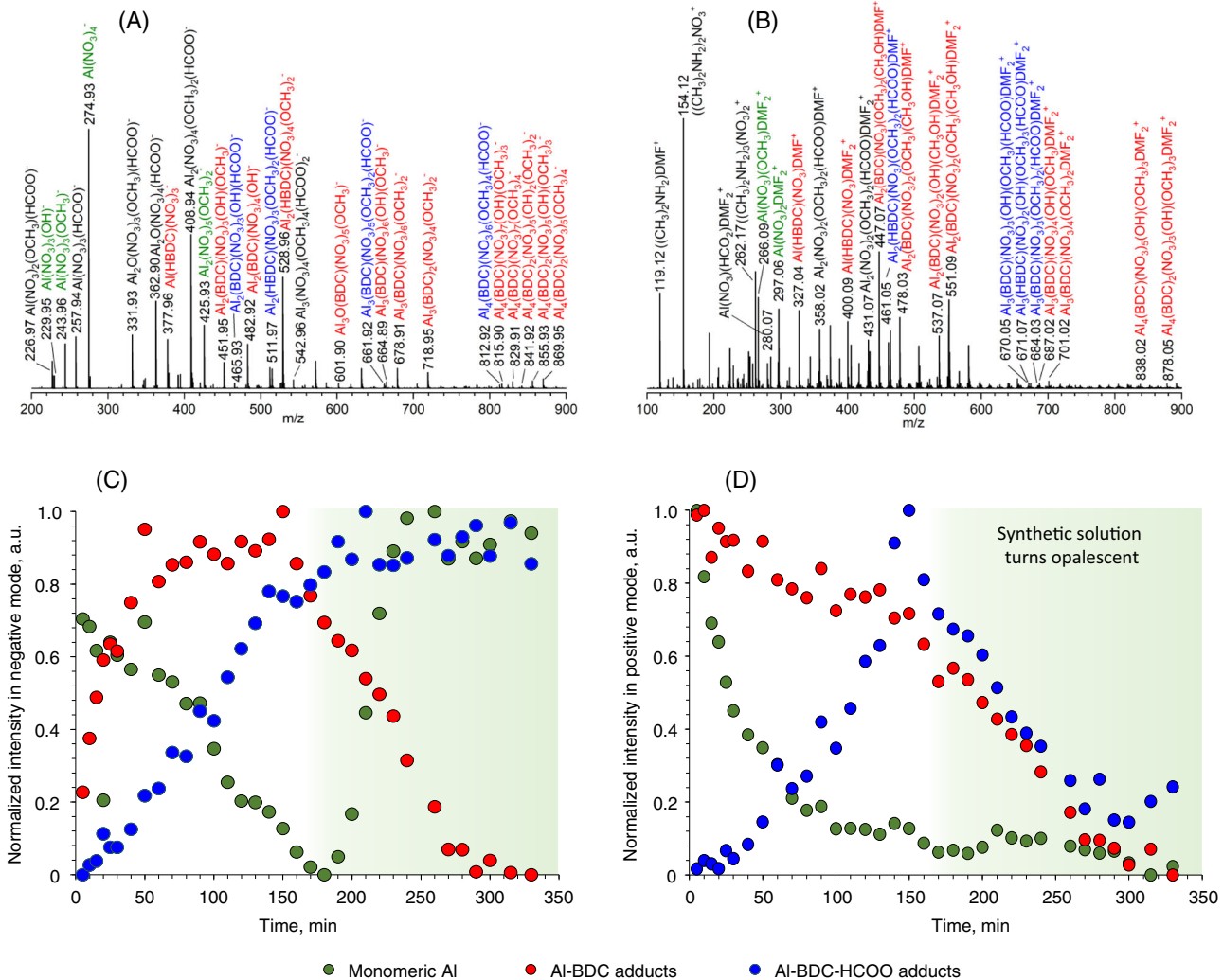

**Fig. 1 ESI Q-TOF mass spectrometry data.** Mass spectra of the synthetic mixture after 100 min measured in (**A**) negative and (**B**) positive modes, respectively; time-resolved evolution of the MS signals attributed to monomeric aluminum, aluminum-terephthalate adducts and aluminum-terephthalate-formate adducts in (**C**) negative and (**D**) positive modes. The reaction was carried out at 343 K, the samples were diluted with methanol (1:100 v/v) prior to injection to mass spectrometer, leading to appearance of methoxy ligands in some of MS peaks. Complete description of the spectra together with the peak assignment is given in Tables S1, S2.

(ii) *Al - BDC* adducts, where the mono- and multinuclear aluminum species coordinated to terephthalic acid were found together with other charge-balancing anions, such as $NO_3^-$, $OH^-$, $OCH_3^-$, and DMF.

(iii) *Al - BDC - HCOO* adducts, which are structurally similar to the previous group, but also possessing formate anions in the structure. Their temporal behavior is significantly different from the other two classes and therefore they were placed in a separate group.

Following this classification, we compared the time-resolved response of the sum of normalized intensities of these main classes (Fig. 1C, D). In the negative mode, the concentration of *Al - BDC* group develops from zero to the certain equilibrated level and after ~150 min of reaction rapidly decreases. Note that the synthetic solution becomes milky at ~150 min of the reaction, indicating the start of MIL-53 formation (see XRD data below). The formate-containing group *Al - BDC - HCOO* slowly progresses during the 0–150 min reaction time and stabilizes. Strikingly, the temporal behavior of *monomeric Al* shows gradual decrease from the beginning of the reaction until the start of crystallization, followed by the rapid increase of the signal and its

stabilization. In contrast, the consumption of cationic *monomeric Al* takes place rapidly, with almost no effect of the crystallization on its concentration. The difference in temporal behavior of anions and cations, however, can not be directly correlated and might originate from the different structure and/or adducts formed. The intensity of the *Al - BDC - HCOO* peaks passes through the maximum at 150 min, while *Al - BDC* group shows consumption of these species during the entire reaction time.

In line with previous reports[30,39], numerous Al- and terephthalic acid-containing species can be identified during the induction period, including those due to the gradual hydrolysis of DMF, namely, dimethylamine and formate anions, with the latter participating in the coordination to aluminum. The species containing formate anions progress slowly, which is seemingly associated with the low hydrolysis rate of DMF. Most of the species containing terephthalic acid are consumed at a longer reaction time, showing its depletion from the solution and transfer to the solid phase. The monomeric aluminum anions show unexpected behavior implying spontaneous increase of the concentration after beginning of MIL-53 crystallization. Even though it can only be interpreted semi-quantitatively, HRMS suggests the presence of such monomeric aluminum anions.

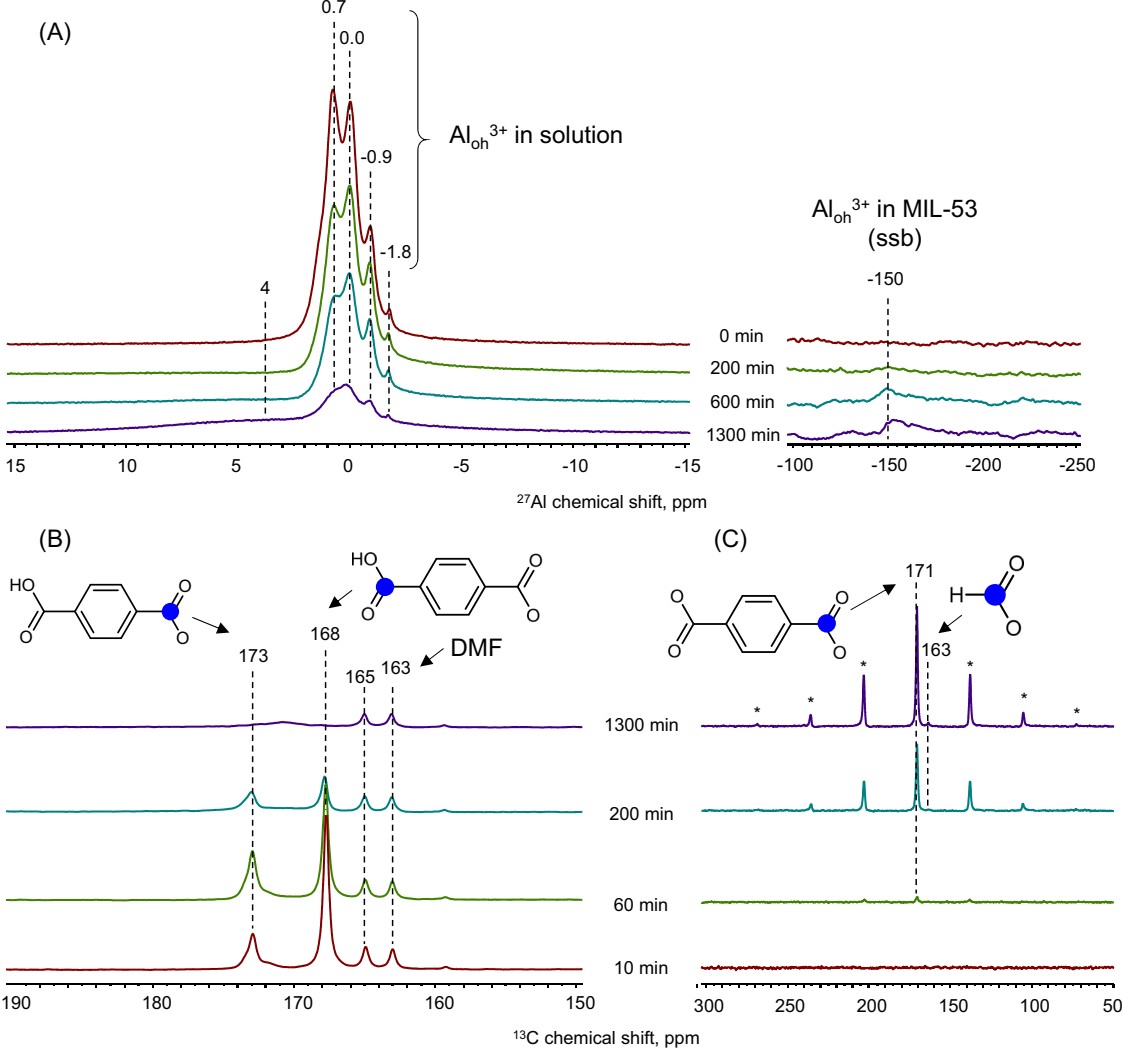

**Fig. 2 Time-resolved MAS NMR data.** Exeplary time-resolved (**A**) $^{27}$Al, (**B**) $^{13}$C direct excitation without proton decoupling, and (**C**) $^{1}$H-$^{13}$C CP-MAS with proton decoupling MAS NMR spectra. The reaction temperature was 343 K, MAS spinning rate varied between 3200 and 3500 Hz.

**In situ MAS NMR spectroscopy**. While HRMS enables[40] establishing the structure of molecular precursors and secondary building units of MIL-53 in solution, MAS NMR can follow[38] the evolution of both liquid and solid phase. Figure 2 presents exemplary $^{13}$C and $^{27}$Al MAS NMR spectra acquired during the synthesis at 343 K. After heating the synthetic mixture to the reaction temperature, several sharp overlapping signals are visible in $^{27}$Al MAS NMR spectrum (Figs. 2A, S5–S7). The signal at 0.0 ppm can be associated with the aluminum atoms in symmetrical octahedral coordination, most probably corresponding to free aluminum nitrate hydrate. Signals at −0.9 and −1.8 ppm are shifted to the lower frequency, indicating the interaction of aluminum with electron-accepting ligands such as carbonyl groups of terephthalic acid. In contrast, the signal at 0.7 ppm most probably originates from the species having a lesser electron-donating ligand than O-doner ligand, N,N-dimethylformamide, in line with HRMS data. The broad signal at 4 ppm can be tentatively assigned to solid products. The left part of the spectrum shows the spinning side bands of a central transition at about 0.0 ppm originating from MIL-53[42]. At the spinning rate used (above 3 kHz), the strongest transition appears at −150 ppm. The time-resolved spectra show the signals around 0 ppm gradually decreasing, and that at −150 ppm developing after some induction period of about 200 min. This points to the consumption of

the aluminum species from the liquid phase and the start of formation of MIL-53.

Figure 2B, C shows complementally $^{13}$C MAS NMR acquired in two modes—direct excitation without proton decoupling and with proton cross-polarization and high-power proton decoupling (CP-MAS), respectively. The latter pulse sequence allows observation of the solid phase, while the former is dominated by the species in solution (Figs. S8–S13). The spectrum of the initial reaction mixture shows signals at 168 and 173 ppm, which are due to the protonated and deprotonated groups of terephthalic acid (Fig. S14). Asymmetry of the peak at 173 ppm might be associated with the contribution of terephthalate bonded to aluminum atoms, in line with HRMS observations. The small signals at 163 and 165 ppm originate from the DMF carboxyl group, undergoing J-splitting on the hydrogen atom. At a longer reaction time about 400 min, the broad signal at 171 ppm becomes visible, which is assigned to terephthalate in the MIL-53 structure[42]. During the reaction, the signal due to the carboxyl groups of terephthalic acid decreases up to complete disappearance pointing to the whole consumption of the latter. In the $^{1}$H-$^{13}$C CP-MAS spectra, no signal is observed at the start of the synthesis. After ~60 min at 343 K, a signal at 171 ppm, accompanied by a spinning side band pattern, appears and develops. This signal is due to the solid species, containing

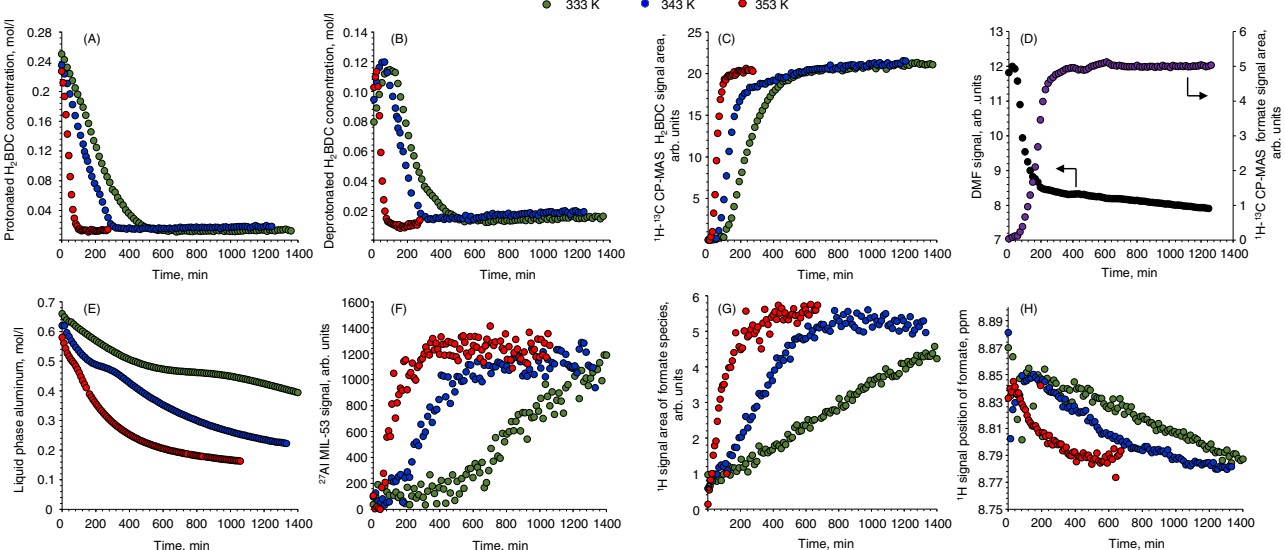

**Fig. 3 Analysis of time-resolved NMR spectra series.** Concentration of (**A**) protonated and (**B**) deprotonated carboxyl group in terephthalic acid obtained from $^{13}$C MAS NMR spectra; temporal evolution of $^{1}$H-$^{13}$C CP-MAS signals of (**C**) terephthalate and (**D**) formate species in solid nuclei of MIL-53 together with the $^{13}$C DMF signal intensity; (**E**) concentration of aluminum species in solution versus synthesis time; (**F**) time-resolved response of the integral intensity of the MIL-53 spinning side band centered at −150 ppm; (**G**) integral intensity and (**H**) position of $^{1}$H signal due to the formate species in solution. Colured circled encode reaction temperature except in (**D**).

terephthalate, which can correspond to the MIL-53 nuclei or phase[42]. Interestingly, another small signal at 163 ppm is visible starting from 200 min, which we tentatively assign to formate species (*vide infra*).

Figure 3A–F shows the kinetic evolution of the discussed signals recorded at 333, 343, and 353 K, derived from the integration of NMR spectra. The concentration of the protonated carboxyl groups of terephthalic acid gradually decreases with the temperature accelerating this process significantly. The complete depletion is observed after 80 min at 353 K and after 600 min at 333 K (Fig. 3A). In contrast, the concentration of deprotonated carboxyl groups of terephthalic acid passes through the maxima, located at the different reaction time with respect to the temperature (Fig. 3B). Such a kinetic behavior points to the intermediate nature of deprotonated terephthalate, suggesting its involvement in the coordination to aluminum rather than direct interaction of $H_2$BDC with aluminum ions. The reaction of deprotonated and protonated carboxyl groups with aluminum is finished simultaneously.

The time-resolved $^{1}$H-$^{13}$C CP-MAS data show the presence of an induction period prior the observation of any signals coming from the solid MIL-53 nuclei within the studied temperature range (Fig. 3C). At 333 K, solid material is observed after 100 min while at 353 K it takes 30 min. Importantly, the start of the formation of solid precursors coincides with the maxima observed for the concentration of the deprotonated terephthalate (Fig. 3B, C), hence emphasizing the key role of the latter for the formation of MIL-53 nuclei. The development of CP-MAS signal finishes simultaneously with the full consumption of terephthalate from the solution, indicating the complete conversion and absence of other forms of terephthalate as side or intermediate products. The temporal behavior of the signal from formate species follows one due to the carboxyl group of terephthalic acid (see Supporting Information). To follow the fate of formate species, we labeled DMF with $^{13}$C in carboxyl group and monitored the evolution of its signal in both liquid and solid phases (Figs. 3D, S15, S16). The decrease of DMF concentration was observed by direct excitation $^{13}$C NMR spectroscopy, confirming the hydrolysis of the solvent. Simultaneously, $^{1}$H-$^{13}$C CP-MAS spectra show that formate

species formed by the hydrolysis of DMF end up in the solid phase of MIL-53 nuclei (Fig. 3D). Moreover, their temporal behavior coincides with the development of the terephthalate species in $^{1}$H-$^{13}$C CP-MAS spectra (Fig. 3C), pointing to the simultaneous inclusion of formate and terephthalate into the MIL-53 nuclei. Figure 3E shows the kinetic behavior of the aluminum species located in the solution, extracted from the integration of the corresponding liquid phase signals. From the beginning of the reaction, the concentration of aluminum decreases to ~0.2 mol/l, which is associated with the excess of aluminum nitrate used in the synthesis. However, the concentration of aluminum does not decrease evenly during the synthesis; instead, a plateau is clearly visible, which occurs at different reaction time depending on the temperature. For instance, at 333 K, after 600 min of synthesis, the concentration of aluminum stabilizes for ~400 min and only then goes down again. Correlating this behavior with the evolution of the solid MIL-53 side band signal (Fig. 3F) shows that the beginning of the plateau in the concentration of soluble aluminum coincides with the start of the formation of the MIL-53 solid phase. This indicates that the initial formation of MIL-53 does not require the continuous transfer of the aluminum from the solution, in contrast, the crystallization occurs via the transformation of pre-formed solid nuclei. Moreover, by comparing $^{27}$Al spectra with $^{13}$C ones, we conclude that the start of the MIL-53 phase formation takes place only after complete consumption of terephthalic acid from the solution and finishing the nucleation process as observed by $^{1}$H-$^{13}$C CP-MAS NMR spectroscopy (Fig. 3A–C). This sequence occurs at all synthesis temperatures.

Apart from the intense signals due to water and N,N-dimethylformamide, $^{1}$H MAS NMR spectra contain the signals corresponding to formate species in solution (see Supporting Information, Figs. S17–S20). Their integral intensities together with the chemical shift values were used to estimate the formate concentration and evolution of pH versus the reaction time (Fig. 3G, H). At the beginning of the reaction, no formate is found in the synthesis mixture; however, the start of the heating induces its slow accumulation via hydrolysis of DMF, catalyzed by the protons formed during deprotonation of terephthalic acid. The

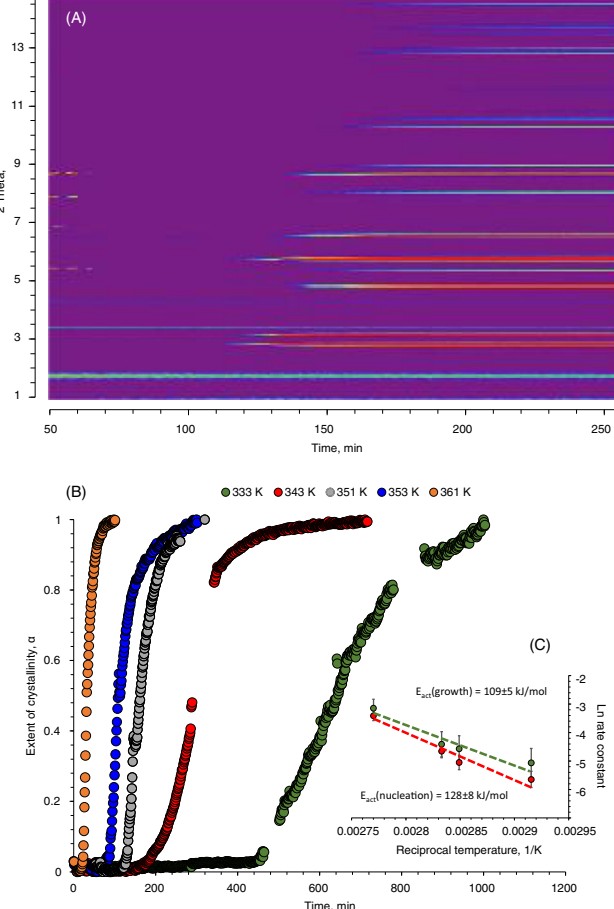

**Fig. 4 Time-resolved X-ray diffraction data.** (**A**) Time-resolved counterplots of XRD data recorded during the synthesis of MIL-53 at 351 K; (**B**) the excess of MIL-53 crystallinity obtained from the background corrected XRD data together with the corresponding fits using Gualtieri model; (**C**) Arrhenius plots for $k_g$ and $k_n$.

formation of formate finishes after the complete crystallization of MIL-53, as evidenced by $^{27}$Al MAS NMR spectra (Fig. 3F). Simultaneously, the position of the formate signal drifts from 8.85 to 8.79 ppm, which indicates the gradual decrease of pH and acidification of the synthetic mixture due to the protons from terephthalic acid. Comparing these data with HRMS, we conclude that formed acid protons interact with dimethylamine, hence leading to a mixture of dimethylammonium nitrate salts of various composition. Moreover, the signals due to methyl groups of these compounds represented by the quadruplet at 35.6 ppm were found in $^{13}$C NMR spectra at long reaction time (Figs. S11–13).

Therefore, the increase in the concentration of deprotonated terephthalate leads to the formation of solid nuclei of MIL-53. After finishing the nucleation phase, accompanied by the complete consumption of terephthalic acid from the solution, concentration of the aluminum in the liquid phase temporarily stabilizes, coinciding with the start of the formation of MOF phase. The hydrolysis of the solvent finishes after the complete crystallization of MIL-53 together with concomitant stabilization of pH.

**In situ X-ray diffraction**. Figure 4A shows the evolution of the XRD reflection during the heating of the synthetic mixture. After induction period, X-ray scattering data shows development of the intense Bragg peaks at $2\Theta = 2.89$, 3.23, and 5.80° that are characteristic of MIL-53 ($I_{mma}$, orthorhombic, $a = 6.9$, $b = 17.6$,

$c = 12.1$ Å)[35,39]. In contrast to the previous report, no other phases such as MOF-235 were found in the course of the synthesis[35]. The obtained background corrected data were used to calculate the excess of crystallinity. Comparing these data with those of $^{27}$Al MAS NMR confirms the compatibility of NMR and XRD: the progression of the MOF phase signals coincides at all temperatures studied (Figs. 3F, 4). However, the insufficient sensitivity of NMR for quadrupolar $^{27}$Al does not allow for the kinetic analysis of the crystallization process.

The analysis of the XRD kinetic profiles was performed using the Gualtieri model (Eq. 1)[43], which enables deconvoluting the nucleation and growth processes. According to the fitted data, the increase of the synthesis temperature affects both the rates of nucleation and crystal growth. The values of kinetic constants show that in the studied temperature range the nucleation is rate-limiting step over the crystal growth, which is in line with the NMR data. The apparent activation energies, extracted from Arrhenius plot correspond to $128 \pm 8$ and $109 \pm 5$ kJ/mol, for nucleation and growth, respectively, which is in line with previously reported values for MIL-53 and MOF-14 synthesis[35,44].

$$\alpha = \frac{1}{1 + e^{-\frac{t-a}{b}}}(1 - e^{-(k_g t)^n}) \tag{1}$$

**Discussion of MIL-53 nucleation and crystallization mechanism**. The formation of MIL-53 starts from establishing the ion equilibrium in the solution (Fig. 5). The terephthalic acid undergoes deprotonation, which is required for coordination to aluminum cations, forming various adducts comprising aluminum, terephthalate, hydroxyl, and nitrate fragments. Remarkably, DMF coordinates and stabilizes exclusively cations in the synthetic mixture. The released protons induce the decrease of pH and hydrolysis of N,N-dimethylformamide to dimethylamine and formate species. The latter interact with aluminum and partially incorporate to aluminum-terephthalate adducts. Dimethylamine is protonated and coordinated to the released nitrate anions, leading to the formation of dimethyamine nitrate salts.

The nucleation process starts after achieving the maximal concentration of deprotonated terephthalate in the mixture (Fig. 5). The formate anions partially incorporate into nuclei with similar to terephthalate rate. The complete consumption of terephthalic acid from the solution terminates nucleation and launches the crystallization of MIL-53 phase. During this stage, the agglomeration of nuclei takes place, accompanied by the release of monomeric species to the solution, hence confirming that solid-solid crystallization mechanism is in action. The formate ions preserve in the MIL-53 product. At the end of crystallization, the hydrolysis of DMF stops together with the stabilization of pH. The main non-crystalline by-product of MIL-53 synthesis is a mixture of soluble dimethylammonium salts with brutto-formula $[((CH_3)_2NH_2)_x(NO_3)_{x-1}]^+[NO_3]^-$.

**Method development and outlook**. Our study on a practical example highlights the capabilities of various time-resolved physical-chemical methods for monitoring the synthesis at different stages and emphasizes the need for a combined approach. It identifies a roadmap towards understanding the chemical processes that occur during the quite underexplored field of synthesis. In particular, NMR spectroscopy shows a great potential for following the fate of molecular precursors, notably organic linkers comprising NMR-active spin $I = 1/2$ nuclei such as $^1$H and $^{13}$C. For the linker concentration, typical for MOF synthesis (about 0.1 M), the time resolution higher than 5 min is achievable with high signal-to-noise ratio. $^1$H-$^{13}$C CP-MAS NMR spectroscopy is ideal for monitoring nucleation phase, both

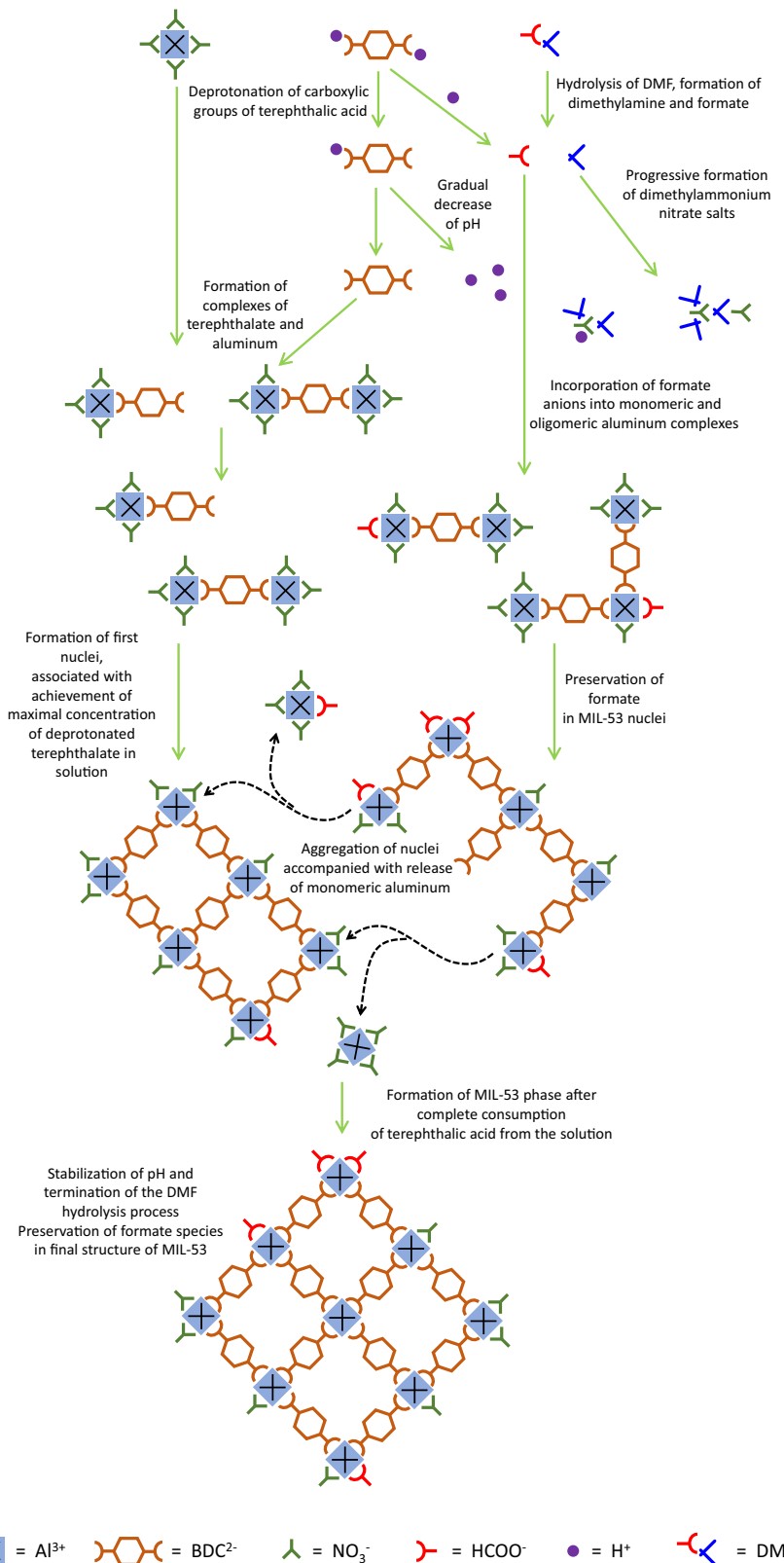

**Fig. 5 Schematic representation of the MIL-53 crystallization mechanism.** The scheme illustrates progressive chemical transformations of molecular precursors leading to the solid MOF product.

kinetics and chemical composition of nuclei. Remarkably, by introducing magic angle spinning, no effect on the synthesis kinetics can be observed: at the spinning rates employed, the local heating due to the viscous friction and centrifugation effect are negligible. However, NMR suffers from low resolution for quadrupolar nuclei such as $^{27}$Al and here very little structural information can be gained, while keeping sufficient sensitivity of the analysis of consumption/accumulation kinetics of precursors and MOF. In this respect, we expect the boost in application of X-ray absorption spectroscopy for monitoring the evolution of

the metal core. Finally, proton NMR was shown to be surprisingly informative in terms of following hydrolysis of the solvent and monitoring pH, even without adding the specific amines, typically used as in situ NMR pH meters.

Equally, we appreciate that NMR is completely blind to the induction phase. We show however that ESI-HRMS can fill this gap by providing structural information about secondary building units and other molecular adducts formed right after the preparation of synthetic mixture. On the one hand, mass spectrometry possesses an almost infinite resolution, dictated by the equipment, and on the other, the deciphering of the spectrum in the region of large masses represents a challenge. However, we show that the isotope labeling allows unambiguous assignment of most observed peaks. Moreover, smart design of the experiments allow extracting (semi)quantitative information, which can be used for kinetic analysis.

Finally, we confirm that the scattering techniques will always go hand in hand with other physical chemical methods, enabling the cross validation of the results and placing them into the context of currently available literature data. Their unique temporal resolution achievable at synchrotrons can not be compared with NMR or HRMS so far, making this data unique in terms of precise kinetic analysis and modeling crystallization.

**Summary**. Critical parts of the mechanism of crystallization of one of the most important aluminum-containing MOFs is revealed. In contrast to what is generally assumed, the crystallization of MIL-53 begins via the aggregation and transformation of pre-formed nuclei rather than via secondary building units and molecular precursors. We show that formate species formed during the hydrolysis of N,N-dimethylformamide partially incorporate in both nuclei and crystals, possibly acting as modulators for the morphology and/or creating defects in the structure. The revealed complexity of the synthesis indicates that unnecessarily simplification and generalization of the mechanisms for the families of MOFs can mislead understanding of their crystallization. In contrast, case studies are highly demanding to explain the experimental observations for each particular MOF. The combination of multiple advanced spectroscopic methods with kinetic measurements is essential to establish any mechanism of nucleation and growth. Our discovery will lead to the creation of a direction in investigation of the synthesis of solids, rejuvenating and boosting activity in this complex field and ultimately enabling the rational design of novel materials.

## Methods

All synthetic mixtures were prepared using aluminum nitrate nonahydrate and terephthalic acid by dissolution in N,N-dimethylformamide (Merck, 99.5%). The final concentrations were 0.66 M and 0.33 M, respectively. For ESI-HRMS and MAS NMR experiments, the $^{13}C_2$-labeled terephthalic acid, $d_7$-DMF, $d_4$-methanol (Sigma-Aldrich, 99%), and $^{15}N$-aluminum nitrate (Cambridge Isotope Laboratories, 98%) were used.

**High-resolution mass spectrometry analysis**. The direct injection analysis of the MOF reaction mixture in the negative and positive ion modes was performed using Agilent 6550 iFunnel Q-TOF high-resolution mass spectrometer, controlled by Agilent MassHunter Workstation Data Acquisition version 10.1. The device was calibrated in positive and negative modes using Agilent ESI-L tuning mix for the mass range 100–3000 $m/z$. Prior to the analysis, the samples where diluted with MeOH (1:100, v/v). To prevent the cross-contamination of the samples, single-use 1 ml plastic syringes were utilized. The flow was set to 5 µl/min. The following mass spectrometer parameters were used: gas temperature: 423 K, drying gas flow: 11 l/min, nebulizer pressure: 10 psi, sheath gas temperature: 473 K, sheath gas flow: 4.5 l/min. The capillary voltage was set to 3000 V. These conditions were selected to avoid severe fragmentation of ions[45]. Mass spectra acquisition time was 1 min. The data analysis was performed using Agilent MassHunter Workstation Qualitative Analysis, version 10.0. The obtained mass spectra were averaged over the acquisition time, and detected peaks and corresponding intensities were extracted as CSV files for further evaluation.

**Magic angle spinning NMR spectroscopy**. The synthesis of MIL-53 was carried out in a sealed glass insert of 5.59 mm outer diameter and ~10 mm length, which was securely placed in the 7-mm zirconia rotor for subsequent in situ NMR analysis of the reaction products. The samples (~150 mg) were placed into the glass tube then kept in liquid nitrogen and flame-sealed.

$^1H$, $^{13}C$, and $^{27}Al$ MAS NMR spectra were recorded on an AVANCE III HD 400WB (Bruker Biospin) spectrometer (9.4 T). A double-resonance 7 mm MAS probe (Bruker Biospin) was employed for detection, the MAS frequency being between 2.8 and 3.6 kHz. VTU unit was used to control the temperature in the stator; the calibration was performed using 80% ethylene glycol in $d_6$-DMSO mixture. Direct polarization without high-power decoupling (NODEC), $^1H$-$^{13}C$ cross-polarization (CP-MAS) and single-pulse $^1H$ and $^{27}Al$ were the main pulse sequences. For $^{13}C$ NODEC, 128 scans were acquired with 4 s recycling delay. For $^1H$ direct excitation, 4 scans were acquired with 5 s recycling delay. For $^{27}Al$ direct excitation, 2048 scans were acquired with 0.25 s recycling delay. In the case of CP-MAS experiments, the CP-RAMP modification with TPPM proton decoupling was used implying accumulation of 128 scans with recycling delay of 3 s and contact time of 5.0 ms. The chemical shifts were calibrated on the $^{13}C$ and $^{27}Al$ spectrum of adamantane and aluminum nitrate as a secondary external reference (38.5 and 0 ppm), respectively. The resulting free induction decays (FIDs) were zero filled and apodized by multiplying with a decaying exponential function with line broadening (LB). After Fourier transform, the signal phasing was manually adjusted and the signal intensity was normalized to the number of scans. To process the data, the TopSpin 3.6 and Mestrenova 10.0 software were applied.

**Time-resolved XRD**. The synthesis of MIL-53 was carried out in a sealed glass capillaries of 2.0 mm outer diameter, 10 µm thickness and ~60 mm length, which was securely placed into the home made cell, enabling continuous heating and spinning of the capillary. Manual goniometer head was used to align the capillary in the beam. The samples (~50 mg) were placed into the glass tube then kept in liquid nitrogen and flame-sealed.

The time-resolved measurements were performed at Material Science beamline at Swiss Light Source, Switzerland, using 25.2 keV radiation[46]. Pilatus 6M detector installed at 50 cm from the capillary was used. After focusing and alignment of the beam, the spinning of the capillary was started together with the heating to the desired temperature with 10 K/min rate. After achieving the desired temperature, the time-resolved series of XRD data were acquired every 1 min until complete crystallization of MIL-53. The data treatment was performed using Medved and Mathematica software[47].

## Data availability

All data needed to evaluate the conclusions are presented in the paper and in Supplementary information file. All RAW data generated during this study are stored on the internal servers of Paul Scherrer Institute and are available from the corresponding authors upon request.

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

## Acknowledgements

We acknowledge the assistance of Mr. Thomas Rohrbach during the adaptation of the synthesis protocol. This project was partially financially supported by the Swiss Innovation Agency lnnoSuisse and is part of the Swiss Competence Centre for Energy Research SCCER BIOSWEET. In addition, part of this work was supported by and performed within the Energy System Integration Platform at the Paul Scherrer Institute.

## Author contributions

V.L.S. devised the idea for the study. D.S. and V.L.S. designed and performed HRMS experiments. S.O.O. and V.L.S. designed the isotope labeling methodology and interpreted HRMS data. N.P.M.C. and V.L.S. performed X-ray scattering experiments and interpreted XRD data. S.B. contributed to the interpretation of HRMS data. R.V. and V.L.S. designed, performed and interpreted NMR experiments. M.R. and J.A.vB. contributed to the design of experiments and discussion of the data. J.A.vB. provided resoures. V.L.S. wrote the draft in close consultation with all the other authors.

## Competing interests

The authors declare no competing interests.
