## [Peer Review File · Nature Communications]

Unraveling the Molecular Mechanism of MIL-53(Al) CrystallizationReviewer #1 (Remarks to the Author):

The manuscript reports the formation mechanism of MIL-53(Al) MOF at the molecular level through combination of three in situ time-resolved analytical tools, namely high-resolution mass spectrometry, NMR spectroscopy and X-ray diffraction. The authors have shown the complementarity of the in situ multi-technique approach for monitoring complex multiscale phenomena, such as crystallization from heterogenous system, by covering both liquid (HRMS, NMR) and solid (MAS NMR, XRD) part. The authors have also shown that the results collected from these various complementary techniques constitute a sufficiently rich data set to provide rational description of the successive chemical events taking place during the synthesis of this MOF. Therefore, the study is interesting and of good quality, but some issues need to be resolved.

1- The authors have proposed a solid-solid transformation mechanism based on a unique observation that is the release of monomeric Al species observed in the HRMS. However, such an argument is not convincing for multiple reasons. As the others themselves said this technique (HRMS) is only semiquantitative. In the absence of absolute quantification, it is difficult to estimate what amount represent the values observed. Moreover, this increase of monomeric Al is observed only in negative mode suggesting therefore the positive corresponding species are negligible. Most importantly, ²⁷Al NMR measurements did not show such sudden monomers increase during the crystallization process. Finally, how to ensure there is no fragmentation in gas phase during the ESI-MS measurements?

2- Although interesting to probe speciation in solution qualitatively, ESI-MS has to be considered with extreme precaution for the reasons mentioned above (non quantitative and destructive technique) but also because of the deviation from real synthesis condition (dilution in methanolic medium). In my opinion, the most informative results come from NMR and in especially ¹³C NMR. However, I do not agree with the assignments proposed. The authors assign the peak at 173 ppm to deprotonated carboxylic function of BDC and the signal at 168 ppm to the same function protonated citing the reference #42, which by the way propose totally different assignment. I suggest alternative assignments for 173 and 168 ppm as due to Al-bounded and free carboxylate groups, respectively. The free carboxylate function should account for both deprotonated and protonated groups in fast chemical exchange with respect to the NMR time scale where their proportion should be dependent on the pH. Based on these assignments, the graph in Fig. 3B shows nicely that crystallization starts when these reactive species reach critical concentration in these supersaturated solutions typically for solution mediated mechanism.

3- In the solid state, the authors assign the signal at 163 ppm to formate species due to DMF hydrolysis. There is no strong evidence of such hypothesis except the ESI-MS observation that again could be due to fragmentation in gas phase measurements, and even though also Hartmann et al. (ref. #39) have showed such hydrolysis in their study. This signal could account for DMF trapped within the pore system of the porous solid during its formation as the signals of methyl groups are seen in the 30-40 ppm range in Figures S8-10. If we assume DMF hydrolysis we should observe beside formate formation dimethyl amine, which should provide single signal of methyl groups (or one quadruplet due to J C-H coupling in non-decoupled ¹³C spectrum), but direct excitation spectra (Fig. S11-13) showed only a couple of signals (2x quadruplet) due to the methyl groups of DMF. There is no evidence of DMF hydrolysis by NMR (neither ¹³C nor ¹H!).

4- Other remarks and suggestions:

a- Lines 72, 97, and 256: "MIL-235" should be corrected to "MOF-235".

b- Line 164: “shifted to the higher frequency, indicating the interaction of aluminum with electron-accepting » should be changed to “shifted to the lower frequency (or higher field), indicating the interaction of aluminum with electron-donating »

c- Line 166: “having an electron-donating ligand » should be changed to « having a lesser electron-donating ligand than O-doner ligand» (see for instance: 10.1016/j.pnmrs.2016.01.003)

d- Line 295 : “NMR -active $S = 1/2$ » should be changed to « NMR -active spin $I = 1/2$ »

e- Line 419: check authors' names and order in ref. #9

f- Fig. 1: Could the authors put the peak assignment labels in A) and B) with the same color code than the color legends for curves in C) and D). (i.e., the monomers in green, Al-BDC adducts in red, and Al-BDC-HCOO adducts in blue)

g- Fig. 3: Authors should add in caption for D) DMF, and could remove null points in B), C), E), and F).

h- In Supporting Information file: Line 102: it should be “from 1 to 5 ms”.

In summary, this work is interesting, but does not fulfil the standards of Nature Communications. I recommend major revision and transfer to another journal as a full paper.

Reviewer #2 (Remarks to the Author):

In this paper Bokhoven and Sushkevich report on the mechanism of formation of MIL-53(Al) using HRMS, MAS solid-state NMR, and X-ray techniques. This paper is very thorough, well written, and important.

With the correction of a few minor typos and language changes (see below), this manuscript is suitable for publication.

I'm sure I missed some minor typos early on, but here are a few that I caught.

Line 127 change 'ot' to 'of'

Line 176 add of for “observation of the solid phase”

Line 244, should probably be “dimethylammonium nitrate” not dimethylamine nitrate

Lastly, the opening line of the conclusions section is bothersome. Line 322 is somewhat overselling and unnecessary given the quality of the work. Stating that “the molecular mechanism” of MIL-53 has been revealed is not entirely accurate. This work shows many components of the mechanism, including how linkers attach themselves to Al-centers via nitrate or formate displacements. However, MIL-53 is an infinite chain, and I don't recall seeing a lot of discussion about how the aluminum chain forms. I would recommend the authors tone down that sentence.

Response to Reviewer 1

The manuscript reports the formation mechanism of MIL-53(Al) MOF at the molecular level through combination of three in situ time-resolved analytical tools, namely high-resolution mass spectrometry, NMR spectroscopy and X-ray diffraction. The authors have shown the complementarity of the in situ multi-technique approach for monitoring complex multiscale phenomena, such as crystallization from heterogenous system, by covering both liquid (HRMS, NMR) and solid (MAS NMR, XRD) part. The authors have also shown that the results collected from these various complementary techniques constitute a sufficiently rich data set to provide rational description of the successive chemical events taking place during the synthesis of this MOF. Therefore, the study is interesting and of good quality, but some issues need to be resolved.

Author's reply.

We thank the Reviewer for the insightful comments and positive assessment of our work. Also we are grateful to the Reviewer for the sharp reading and critical comments, which helped us to improve the presentation, remove inconsistencies, and clarify data interpretation.

The authors have proposed a solid-solid transformation mechanism based on a unique observation that is the release of monomeric Al species observed in the HRMS. However, such an argument is not convincing for multiple reasons. As the others themselves said this technique (HRMS) is only semiquantitative. In the absence of absolute quantification, it is difficult to estimate what amount represent the values observed. Moreover, this increase of monomeric Al is observed only in negative mode suggesting therefore the positive corresponding species are negligible. Most importantly, 27Al NMR measurements did not show such sudden monomers increase during the crystallization process. Finally, how to ensure there is no fragmentation in gas phase during the ESI-MS measurements?

Author's reply.

This question is very important. Indeed, under the utilized conditions HRMS is semiquantitative and can not speak on its own about the release of the monomeric aluminum, not bound to terephthalate, after the start of crystallization. However, in our discussion we did not rely solely on the HRMS results. Originally, the kinetic behavior of the 27Al NMR signal pointed out that the consumption of Al from the solution does not proceed evenly and continuously, and that there are temporal periods, when the concentration of Al stabilizes (Figure 3 E). Crucially, this was observed for every temperature studied. Such a kinetic behavior could be associated either with the complete termination of Al consumption from the solution (for which it is difficult to imagine a mechanism) or with the release of some Al from the solidified species. The second hypothesis found its support by the HRMS data. However, the exact amount of released Al can not be directly measured due to the reasons discussed above. We continue working on this problem, trying to elaborate the methodology, which enables quantitative measurements using internal standards of appropriate structure and m/z range. That the kinetic behavior of the 27Al NMR signal is at the origin of the model and not the semi-quantitative HRMS is emphasized in the text by adding the following sentence on the Page 8.

In general, one should not expect similar or identical temporal behavior of the positively and negatively charged ions, even of a similar structure. Moreover, the species assigned to the monomeric aluminum in our case have different structures in the negative and positive ionization modes. In particular, the ions observed in the positive ionization mode were detected as DMF adducts, while this was not the case for the ions observed in the negative ionization mode. Therefore, we do not think that the observation of different temporal behavior in negative and positive ionization modes can be unambiguously interpreted or correlated to each other. To highlight that issue, the discussion has been added on the Page 7.

Concerning the fragmentation of the compounds during the ESI-HRMS measurements, we utilized the

ionization conditions with relatively low gas temperature and capillary voltage to avoid severe fragmentation. This approach was previously confirmed to be efficient and neat for other MOFs (Ref. 45). We, therefore, do not expect a significant contribution of the fragmentation to the obtained data and conclusions made. An explaining sentence has been added to the text on Page 16.

Action taken.

A brief discussion has been added on the Page 8:

“The monomeric aluminum anions show unexpected behaviour implying spontaneous increase of the concentration after beginning of MIL-53 crystallization. Even though it can only be interpreted semi-quantitatively, HRMS suggests the presence of such monomeric aluminum anions.”

Detailed experimental protocol and reference have been included to the Materials and Methods section on the Page 16: “The following mass spectrometer parameters were used: gas temperature : 423 K, drying gas flow : 11 l/min, nebulizer pressure : 10 psi, sheath gas temperature : 473 K, sheath gas flow : 4.5 l/min. The capillary voltage was set to 3000 V. These conditions were selected to avoid severe fragmentation of ions.⁴⁵”

Explanation of the origin of different temporal behavior for cations and anions has been added on the Page 7: “The difference in temporal behavior of anions and cations, however, can not be directly correlated and might originate from the different structure and/or adducts formed.”

Although interesting to probe speciation in solution qualitatively, ESI-MS has to be considered with extreme precaution for the reasons mentioned above (non quantitative and destructive technique) but also because of the deviation from real synthesis condition (dilution in methanolic medium). In my opinion, the most informative results come from NMR and in especially ¹³C NMR. However, I do not agree with the assignments proposed. The authors assign the peak at 173 ppm to deprotonated carboxylic function of BDC and the signal at 168 ppm to the same function protonated citing the reference #42, which by the way propose totally different assignment. I suggest alternative assignments for 173 and 168 ppm as due to Al-bounded and free carboxylate groups, respectively. The free carboxylate function should account for both deprotonated and protonated groups in fast chemical exchange with respect to the NMR time scale where their proportion should be dependent on the pH. Based on these

assignments, the graph in Fig. 3B shows nicely that crystallization starts when these reactive species reach critical concentration in these supersaturates solutions typically for solution mediated mechanism.

Author's reply.

This is a valid point. To independently verify the assignment of the ¹³C NMR signals, two more samples were measured, in particular, ¹³C-labelled terephthalic acid in i) 1M solution of HCl in DMF and in ii) 1M Et₃N in DMF. These two solutions mimic the acidic and basic media, shifting the protonation-deprotonation equilibrium for terephthalic acid. Obviously, in HCl/DMF mixture terephthalic acid will be fully protonated, while in Et₃N/DMF it should be completely deprotonated. The corresponding spectra show that protonated terephthalate possesses the signal at 172 ppm, while deprotonated has the signal at 168 ppm. This suggests the correct original assignment of the major peaks in ¹³C NMR spectra. The origin of such behavior in light of high tendency of carboxyl groups to a fast chemical exchange is still under exploration.

However, we also agree with the Reviewer, that the signal around 173 ppm is not a single peak and shows the presence of a shoulder and possibly other small-intensity signals. The most reasonable origin of these peaks relates to Al-coordinated terephthalate, which was also observed using HRMS. In light of that, the assignment and discussion of NMR data has been adjusted accordingly. The following changes have been made together with the addition of a figure showing the NMR spectra of HCl/DMF and Et₃N/DMF mixtures in the Supporting Information.

Action taken.

On the Page 9 of main text the following text has been added:

“The spectrum of the initial reaction mixture shows signals at 168 and 173 ppm, which are due to the protonated and deprotonated groups of terephthalic acid (Figure S14). Asymmetry of the peak at 173 ppm might be associated with the contribution of terephthalate bonded to aluminum atoms, in line with HRMS observations.”

In the solid state, the authors assign the signal at 163 ppm to formate species due to DMF hydrolysis. There is no strong evidence of such hypothesis except the ESI-MS observation that again could be due to fragmentation in gas phase measurements, and even though also Hartmann et al. (ref. #39) have showed such hydrolysis in their study. This signal could account for DMF trapped within the pore system of the porous solid during its formation as the signals of methyl groups are seen in the 30-40 ppm range in Figures S8-10. If we assume DMF hydrolysis we should observe beside formate formation dimethyl amine, which should provide single signal of methyl groups (or one quadruplet due to J C-H coupling in non-decoupled ¹³C spectrum), but direct excitation spectra (Fig. S11-13) showed only a couple of signals (2x quadruplet) due to the methyl groups of DMF. There is no evidence of DMF hydrolysis by NMR (neither ¹³C nor ¹H!).

Author's reply.

This is a good question, we thank the Reviewer for raising it. Indeed, one should expect the formation of dimethylamine together with the formate species. Practically, we do observe dimethylamine in ¹³C NMR; there is a clear set of NMR signals, which are separated from ones due to dimethylamine. They are not resolved in the original submission due to erroneously applied apodisation. The ¹³C NMR spectra without apodization have now been included to the Supporting Information. In the proton NMR, we indeed do not observe any signals due to dimethylamine, apparently due to the broadening and/or severe overlapping with other signals, coming from DMF.

Action taken.

Following this discussion, we have modified the text on the Page 11: “Moreover, the signals due to methyl groups of these compounds represented by the quadruplet at 35.6 ppm were found in ¹³C NMR spectra at long reaction time (Figures S11-13).”

Other remarks and suggestions:

a- Lines 72, 97, and 256: “MIL-235” should be corrected to “MOF-235”.

b- Line 164: “shifted to the higher frequency, indicating the interaction of aluminum with electron-accepting » should be changed to “shifted to the lower frequency (or higher field), indicating the interaction of aluminum with electron-donating »

c- Line 166: “having an electron-donating ligand » should be changed to « having a lesser electron-donating ligand than O-donor ligand» (see for instance: 10.1016/j.pnmrs.2016.01.003)

d- Line 295 : “NMR -active S = 1/2 » should be changed to « NMR -active spin I = 1/2 »

e- Line 419: check authors' names and order in ref. #9

Author's reply.

The requested changes have all been implemented.

f- Fig. 1: Could the authors put the peak assignment labels in A) and B) with the same color code than the color legends for curves in C) and D). (i.e., the monomers in green, Al-BDC adducts in red, and Al-BDC-HCOO adducts in blue)

g- Fig. 3: Authors should add in caption for D) DMF, and could remove null points in B), C), E), and F).

Author's reply.

The corresponding Figures have been modified accordingly.

h- In Supporting Information file: Line 102: it should be "from 1 to 5 ms".

Author's reply.

The typo has been corrected.

Response to Reviewer 2

In this paper Bokhoven and Sushkevich report on the mechanism of formation of MIL-53(Al) using HRMS, MAS solid-state NMR, and X-ray techniques. This paper is very thorough, well written, and important. With the correction of a few minor typos and language changes (see below), this manuscript is suitable for publication.

Author's reply.

We thank the Reviewer for a positive assessment of our work and helpful comments, which we address below.

I'm sure I missed some minor typos early on, but here are a few that I caught.

Line 127 change 'ot' to 'of'

Line 176 add of for "observation of the solid phase"

Line 244, should probably be "dimethylammonium nitrate" not dimethylamine nitrate

Author's reply.

The typos have been corrected.

Lastly, the opening line of the conclusions section is bothersome. Line 322 is somewhat overselling and unnecessary given the quality of the work. Stating that "the molecular mechanism" of MIL-53 has been revealed is not entirely accurate. This work shows many components of the mechanism, including how linkers attach themselves to Al-centers via nitrate or formate displacements. However, MIL-53 is an infinite chain, and I don't recall seeing a lot of discussion about how the aluminum chain forms. I would recommend the authors tone down that sentence.

Author's reply.

We agree with this comment and the tone on the final conclusions has been changed accordingly on the Page 15:

"Crucial parts of the mechanism of crystallization of one of the most important aluminum-containing MOFs is revealed."

Reviewer #1 (Remarks to the Author):

The authors have responded satisfactorily to all questions and comments raised by the reviewers.